# Comparative Study of Digital Breast Tomosynthesis (DBT) with and without Ultrasound versus Breast Magnetic Resonance Imaging (MRI) in Detecting Breast Lesion

**DOI:** 10.3390/ijerph19020759

**Published:** 2022-01-11

**Authors:** Janice Hui Ling Goh, Toh Leong Tan, Suraya Aziz, Iqbal Hussain Rizuana

**Affiliations:** 1Department of Radiology, Faculty of Medicine, Universiti Kebangsaan Malaysia, Cheras 56000, Malaysia; janicegoh2016@gmail.com (J.H.L.G.); suraya.aziz@gmail.com (S.A.); 2Department of Emergency Medicine, Faculty of Medicine, Universiti Kebangsaan Malaysia, Cheras 56000, Malaysia; sebastianttl@yahoo.co.uk

**Keywords:** digital breast tomosynthesis, DBT, 3D mammography, breast MRI, ultrasonography

## Abstract

Digital breast tomosynthesis (DBT) is a fairly recent breast imaging technique invented to overcome the challenges of overlapping breast tissue. Ultrasonography (USG) was used as a complementary tool to DBT for the purpose of this study. Nonetheless, breast magnetic resonance imaging (MRI) remains the most sensitive tool to detect breast lesion. The purpose of this study was to evaluate diagnostic performance of DBT, with and without USG, versus breast MRI in correlation to histopathological examination (HPE). This was a retrospective study in a university hospital over a duration of 24 months. Findings were acquired from a formal report and were correlated with HPE. The sensitivity of DBT with or without USG was lower than MRI. However, the accuracy, specificity and PPV were raised with the aid of USG to equivalent or better than MRI. These three modalities showed statistically significant in correlation with HPE (*p* < 0.005, chi-squared). Generally, DBT alone has lower sensitivity as compared to MRI. However, it is reassuring that DBT + USG could significantly improve diagnostic performance to that comparable to MRI. In conclusion, results of this study are vital to centers which do not have MRI, as complementary ultrasound can accentuate diagnostic performance of DBT.

## 1. Introduction

Breast cancer is the most common cancer in women, especially among middle-aged and older women [1,2,3]. Mammography, introduced in the late 1990s is universally used as a screening tool to detect early breast cancer. It is relatively inexpensive and easily available. It plays a vital role in early detection of breast cancer where the changes may be picked up as early as two years on mammogram before it becomes symptomatic [2]. Screening mammography can be carried out starting at the age of 30 for women in the high-risk group [3]. Early detection provides the best opportunity for disease remission. Research shows that annual mammograms for early breast cancer detection reduces mortality up to 30% in screened women [4]. Therefore, early breast cancer detection is encouraged amongst all women, especially for women from ages 45 and above. Although mammography is the imaging of choice for breast cancer screening with a sensitivity ranging from 70 to 85%, it is reduced to 30–50% in women with dense breasts because a dense breast can mask an underlying lesion [5].

Digital breast tomosynthesis (DBT) was introduced to overcome challenges in breast imaging. It is a fairly recent imaging technique, which is computed in 3 dimensions and enables separation of tissue planes to overcome the challenges of overlapping breast tissue. This 3-dimensional mammography imaging captures multiple two-dimensional breast images and volumetrically reconstructs them into high-quality three-dimensional images, allowing images of the breast to be reviewed in the form of “slices”, similar to those of a computerized tomography scan (CT-scan). This offers an advantage for dense breasts with overlapping tissue where there will be less superimposed structures, thus aiding tremendously in early cancer detection and therefore improving the prognosis amongst breast cancer patients [5].

DBT has been proven to have more benefits compared to conventional two-dimensional (2D) mammography in terms of cancer detection. Many studies have been conducted to compare the performance between 2D mammography and DBT. A recent retrospective study by McDonald et al., has proven that DBT screening significantly reduced the recall rates and increased cancer detection [6]. Zuckerman et al., (2016) further supported this finding in their multi-center cohort study [7]. DBT, as a recent development in mammographic imaging procedures, is still undergoing extensive research. There are many studies which compared 3D mammography with MRI. However, these studies yield variable results. Roganovic et al. and Kamal et al, reported that DBT has a sensitivity of 92 to 93%, which is comparable to breast MRI (92 to 100%) [8,9]. Saxena et al., reported that MRI has better specificity compared to DBT [10]. However, Comstock et al., shows that DBT has sensitivity as low as 39% but with specificity as high as 97%, which both showed significant difference compared to MRI [11].

The sensitivity of mammography is substantially affected by the radiographic density of the breasts. Breast lesions, especially those with no calcified component, are likely to be overlooked in women with dense breasts, resulting in missed cancers on mammography. Therefore, USG of the breasts is often used as adjunctive to mammography, especially in women with radiographically dense breast, to increase sensitivity in detecting breast lesions. This was supported by multiple studies done in the past decades, which have shown that 2D mammography with supplementary ultrasonography increased cancer detection rate [12,13,14,15,16,17,18].

Breast MRI has been shown to be the most sensitive tool to detect breast cancer with a reported sensitivity of MRI for visualization of invasive cancer reaching up to 100%, as well as its ability to demonstrate clinically, radiographically, and sonographically occult breast cancer [19,20]. However, despite its superior diagnostic value, there are limitations in widespread MRI usage due to its cost, availability and protracted scanning time. MRI is also unsuitable for patients on pacemakers, patients with non-MRI compatible aneurysm clips/metallic implants, or severe claustrophobia [19,20].

Although multiple studies have proven that diagnostic performance of 2D mammography improved with adjunctive USG, there have been limited studies conducted to evaluate diagnostic performance of DBT with USG in comparison to MRI. Therefore, for the purpose of this study, three modalities were used, DBT, DBT with USG, and MRI, correlating to the histopathological result which was taken as the gold standard. If our study is able to prove that the diagnostic performance of DBT with or without USG is comparable to MRI, then a cheaper modality would be an alternative to MRI.

## 2. Materials and Methods

The study was approved by the Ethics Committee of Universiti Kebangsaan Malaysia (JEP-2020-614, 3 August 2020). This is a retrospective descriptive study. Thus, no informed or written consent was required.

A retrospective study was conducted in the radiology department of a university hospital for a period of 24 months. The study protocol was approved by the institutional local ethics committee. Images were obtained through the hospital database. During these 24 months, a total of 3747 cases of DBT with USG were done for screening and diagnostic purposes. Forty-seven female patients out of the 3747 had their breast MRI done. Their ages ranged from 36 to 78 years old, with mean age 55.14 years. Their ethnicity was 72% Malay (*N* = 10), 21% Chinese (*N* = 3) and 7% Indian (*N* = 1).

According to the inclusion criteria, only patients who underwent all 3 modalities, DBT, USG and MRI were included. Histopathological examination (HPE) was taken as a gold standard for reference. In the absence of HPE, a lesion that was stable for at least 12 months, was considered as a benign lesion for the purpose of our study. Among the 47 patients with DBT, USG and MRI, 33 were excluded due to lack of HPE or follow-up (*N* = 12), absence of DBT or USG images (*N* = 11), breast implant-related pathology (e.g., implant rupture) (*N* = 7) and lesion was surgically removed in between the imaging (*N* = 3). Finally, out of all these, only 14 patients fulfilled our inclusion criteria. From these 14 patients, 24 breast lesions were identified. Summary of the above patients selection is shown in Figure 1. 

### 2.1. Imaging Protocols

DBT *was* performed using a dedicated digital mammography u*n*it (Selenia Dimensions; Hologic, Bedford, MA, USA) with a selenium (Se) detector 24 × 29 cm and aluminium (Al) filter. Standard projections, which were mediolateral oblique (MLO), and craniocaudal (CC) views were taken with 34–49 peak kilovoltage (kVp) and maximum 200 milliampere-seconds (mAs). Images were reconstructed into 1 mm slice thickness with reconstruction time of 5 s.

All DBT examination was supplemented with ultrasonography (Affiniti 50; Phillips, United States of America) using a high-resolution L18-5 transducer with patient in supine position. Transverse and longitudinal images of breast lesion was obtained.

MRI breast was performed using 1.5T scanner (Magnetom Avanto fit; Siemens, Erlangen, Germany). Patient was placed in prone position with dedicated breast surface coil. Imaging was done with standard imaging protocols, which were bilateral T1-weighted, T2-weighted, Short Tau Inversion Recovery (STIR), T2 fat saturated sequences with diffusion-weighted imaging/apparent diffusion coefficient (DWI/ADC), Spectral Adiabatic Inversion Recovery (SPAIR) and FL-3D. This was followed by dynamic contrast study with total of 6 dynamic acquisitions, one before and 5 after an intravenous contrast media (Gadovist, Bayer Schering Pharma, Berlin, Germany), based on 0.1 mmol/kg body weight, in slice thickness of 1 mm and acquisition matrix of 336 × 448. The lesions were reviewed for their morphology and kinetic curves.

All the data from the three modalities were transferred to a workstation for analysis and interpretation.

### 2.2. Image Interpretation

Findings of breast lesions on DBT, USG and MRI were acquired from formal report by of 5 radiologists with 2 years of DBT imaging experience in our center. All lesions were scored according to American College of Radiology (ACR) Breast Imaging-Reporting Data and System (BI-RADS) 5th Edition lexicon criteria, and were compared to the HPE result, whenever feasible. For lesions which did not have HPE results, at least 12 months of lesion stability was taken as benign lesion. BI-RADS categories 1 and 2 were considered benign, while categories 3 to 5 were considered malignant.

## 3. Results

There were 14 patients included in the study as summarized in Table 1. Their age ranged from 36 to 78 years old, with a mean age of 55.14 years. Their ethnicity was 72% Malay (*N* = 10), 21% Chinese (*N* = 3) and 7% Indian (*N* = 1). Eight of the patients were symptomatic (e.g., presence of breast lump, inverted nipple), while 6 of them were for screening (e.g., for breast carcinoma follow-up, on hormonal therapy and to look for primary malignancy).

The intervals between the three imaging modalities were variable. One patient had all imaging done on the same day, while 3 patients had imaging intervals between 1 to 3 months, 5 patients had imaging done in less than a month, and another 5 patients had imaging intervals of more than 3 months.

Twenty-four lesions which fulfilled the criteria are summarized in Table 2. Out of the 24 lesions, 9 lesions had malignant HPE (4 intralobular carcinoma and 5 invasive ductal carcinoma), whereas 15 lesions were benign. Among the 15 benign lesions, 5 were benign breast tissue, 1 was fibroadenoma and 9 lesions had either resolved or were stable on follow-up.

The breast composition was divided according to the American College of Radiology (ACR) BI-RADS lexicon. The majority of patients had breast density c, which comprised 50% (*N* = 7), followed by density b 36% (*N* = 5) and 7% for both density a and d (*N* = 1), as shown in Figure 2.

BI-RADS density c was seen across all age groups, with highest among the age group of 41 to 50 years. BI-RADS density b was seen in patients above the age of 51 years old. BI-RADS density a was only seen in patients aged 51–60 years, and BI-RADS density d was only seen in patients aged 31–40 years old age groups in our study.

Comparison of DBT, DBT + USG and MRI with HPE result or lesion stability on follow-up is shown in Table 3. Both DBT + USG and MRI identified all the benign lesions (true negative). For malignant lesions (true positive), both DBT and DBT + USG identified 88.89% while MRI depicted 100% of malignant lesions. DBT has 26.67% false positive, whereas DBT + USG and MRI have 20%. Both DBT and DBT + USG have 11.11% false negative, whereas MRI does not identify any false negative lesion.

Table 4 shows standard indices for all three modalities. Accuracy for DBT, DBT + USG and MRI ranges 79.1% to 87.5%. Sensitivity and specificity for DBT, DBT + USG and MRI ranges from 88.9% to 100%, and 73.3% to 80%, respectively.

Positive predictive value (PPV) for breast lesions in DBT, DBT + USG and MRI ranges from 66.7% to 75%. Negative predictive value (NPV) DBT, DBT + USG and MRI ranges from 91.6% to 100%.

The correlation between BI-RADS category of DBT and HPE, DBT + USG and HPE, as well as MRI and HPE yielded *p*-value of <0.005 which means significant differences in all three modalities (*p* < 0.005, Chi-square), as summarized in Table 5.

## 4. Discussion

Several studies have been conducted to evaluate diagnostic performance of DBT, DBT + USG and MRI separately. However, to the best of our knowledge, there was no study conducted to evaluate all these modalities simultaneously in the English literature. Therefore, present study was intended to evaluate and compare diagnostic performance in all three modalities, which are DBT, DBT + USG, and MRI, correlating with HPE result which was taken as gold standard. Table 6 showed comparison of the present study with other studies.

Our study had a slightly higher number of symptomatic patients compared to asymptotic cases because our center is a tertiary referral center. The present study regarded BI-RADS 1 and 2 as benign, while BI-RADS 3 to 5 were malignant. BI-RADS 3 lesion is categorized as “probably benign” with ≤2% likelihood of cancer in ACR BI-RADS definition that requires follow-up assessment. However, recent studies for BI-RADS 3 lesion with histopathological confirmation showed that up to 34% is malignant [21,22]. A recent systemic review and meta-analysis which involved data from 129 studies also stated higher incidence of malignancy rate in BI-RADS 3 lesions, which was 17% [23]. Therefore, our study categorized BI-RADS 3 lesion as malignant so as to not miss any potentially malignant lesions.

Systemic review of 2D mammography supplemented with screening USG showed improvement in median cancer detection rate of 4.2 (0.3–7.7) [16]. A study done by Sudhir et.al on contrast-enhanced digital mammography (CEDM) where they also included DBT and DBT + USG showed that there was a fair increased in sensitivity, which were 82.8% and 88.5% respectively [24]. Another study done by Dibble et al. assessing supplementary USG to both digital mammography and DBT showed a slight increase in cancer detection rate; however, there was no significant difference [25]. A study combining DBT with and without USG Doppler had demonstrated a slight increase in sensitivity from 98.6% to 100% [26]. Mariscotti et al. conducted a study involving both 2D and 3D mammography supplemented with USG compared to MRI of the breast, where 2D and 3D mammography with adjunct USG had comparable sensitivity with MRI, which were 97.7% versus 98.8%. The overall accuracy of 2D mammography with MRI compared to DBT with USG were also equivalent, which were 92.3 and 93.7%, respectively [27].

DBT, a new technology that acquires multiple low-dose mammographic projections through the breast is approved for use in the United States despite its added radiation dose. This compromise was made due to promising early data in which DBT was proven to reduce recall rate and improve cancer detection rate. There was even reduction in noncancer recall rates, thus preventing unnecessary additional diagnostic testing in women with benign lesions [28,29]. A recent multi-institutional study found that DBT was able to improve characterization of invasive cancers by making masses, asymmetries, and architectural distortion more apparent [28,29]. Enhanced diagnostic capabilities of DBT is presumably attributed to its ability to reduce breast tissue superimposition [28].

Currently, only a few centers in the U.S. have DBT. If DBT lives up to its potential to reduce recall rate while increasing cancer detection rate, it will likely become the next digital mammography.

In our study, all the benign lesions were identified on MRI, but only 5 of these lesions were visualized in DBT (33.33%). With complimentary ultrasonography, a total 10 lesions could be identified (66.67%). Patients with benign lesions that were not visualized on DBT had dense breasts, i.e., BI-RADS composition c and d. Förnvik et al. studied the diagnostic performance of DBT in women with dense breasts reported that breast composition can be associated with false negative on DBT, although not statistically significant [30]. This was also supported by another study done by Mariscotti et al. [27].

Our study showed that accuracy for DBT, DBT + USG and MRI were 79.1%, 83.3% and 79.1%, respectively. This was in accordance with a study done by Sudhir et al., whose study showed similar accuracy of 83.7% for DBT and 84.3% for DBT + USG; however, it was not statistically significant [24]. Saxena et al., on the other hand reported better accuracy for DBT and MRI, which were 90% and 98%, respectively [10]. This could be due to the differences in lesion categorization between these studies; in their study, BI-RADS 4 and 5 were regarded as malignant, whereas in our study BI-RADS 3, 4 and 5 were regarded as malignant.

The sensitivity of DBT and DBT + USG were equivalent, which was 88.89%, whereas it was 100% on MRI. Our finding was comparable to several studies which reported sensitivity of DBT ranging from 90–93% with sensitivity of MRI near 100% [9,10,11]. In a study done by Kamal et al., in which they assessed the ability of DBT and MRI in improving the sensitivity of digital mammography, they noted that DBT alone had equal sensitivity compared to MRI, which was 92% [9]. This finding is also supported by another study in which they reported that both DBT and MRI has sensitivity of 100% [8]. On the contrary, Comstock et al., who studied DBT with abbreviated MRI showed that sensitivity of MRI was 95.7%, but the sensitivity of DBT in detecting breast cancer was significantly lower (39.1%). This could be due to the fact that their study population only included women with dense breasts with a density of c and d [11]. Mariscotti et.al, who studied mammography with adjunctive USG reported that sensitivity of DBT alone, which was 90.7%, increased to 97.7% when combining all 3 modalities (DM + DBT + US). However, they concluded that digital mammography combined with DBT and USG was as good as MRI because the overall accuracy of these techniques was 93.7% and 92.3%, respectively [27].

Although the sensitivity of DBT with and without USG was not as good as MRI, the specificity for DBT alone was better than MRI, which was 73.3% versus 66.7%, and was further increased with a supplementary USG to 80% in our study. These findings were in accordance with Saxena et al., who showed specificity of DBT alone was 77.2%. However, their specificity of MRI was higher, at 95.5%, and it showed statistical significance [10]. This could be because our study regarded BI-RADS 3 lesion as malignant. In another study, specificity for MRI was lower, only 60.7%, due to their false positive findings, e.g., intramammary lymph node, physiological or post-operative changes [8].

The PPV of all three modalities in our study were generally lower, which were 66.7% for DBT, 72.7% for DBT + USG and 64.3% for MRI, as compared to other literature review, i.e., 80.6–85.7% for DBT, 82.8% for DBT + USG and 71.1–96.6% for MRI [8,10,24]. This was likely due to the nature of lesion categorization in our study, i.e., BI-RADS 3 was categorized as malignant.

NPV values in our study for all three modalities were less variable compared to other studies, which were 91.6%, 92.3% and 100% versus 81.7–100%, 86.3% and 89.5–100% for DBT, DBT + USG and MRI, respectively [8,10,24].

The MRI machine itself is extremely expensive and could cost double the price of a DBT machine. In our local setting, a patient would need to pay 4 times the price of a DBT + USG examination for an MRI examination. Furthermore, MRI also has a long image acquisition time with an estimated duration of at least 30 min, compared to DBT, which requires less than 8 seconds for each projection and a total of 16 seconds for both MLO-CC projections in our institute. DBT can also be reliably performed in a more comfortable patient setting as compared to MRI, for example, in patients with claustrophobia, who are not able to undergo an MRI scan.

Overall, our study showed DBT alone had lower sensitivity as compared to MRI; however, it was better in specificity and comparable to MRI in terms of accuracy and PPV. With the aid of USG, its accuracy, specificity and PPV could be further enhanced. This was shown when accuracy increased from 79.1% to 83.3%, specificity increased from 73.3% to 80%, and PPV increased from 66.7% to 72.7% for DBT supplemented with USG.

## 5. Limitations

There was a variability in the examination intervals between images taken on the same day to those taken at intervals of more than 3 months due to the lockdown period during the COVID-19 pandemic. The current data has a limited sample size; therefore, we plan to conduct a multi-center study incorporating data from other centers to extend our results into a larger sample size in the near future.

## 6. Conclusions

Our study demonstrated that DBT alone has a fair sensitivity in lesion diagnosis; however, it is quite reassuring that DBT with complementary ultrasound has significantly improved diagnostic performance, comparable to that of MRI. Therefore, results of this study are vital to centers which do not have the luxury of having an expensive MRI machine or breast radiologists to interpret MRI images. This study also increases confidence of general radiologists where complementary ultrasound can enhance their diagnostic performance for DBT.

## Figures and Tables

**Figure 1 ijerph-19-00759-f001:**
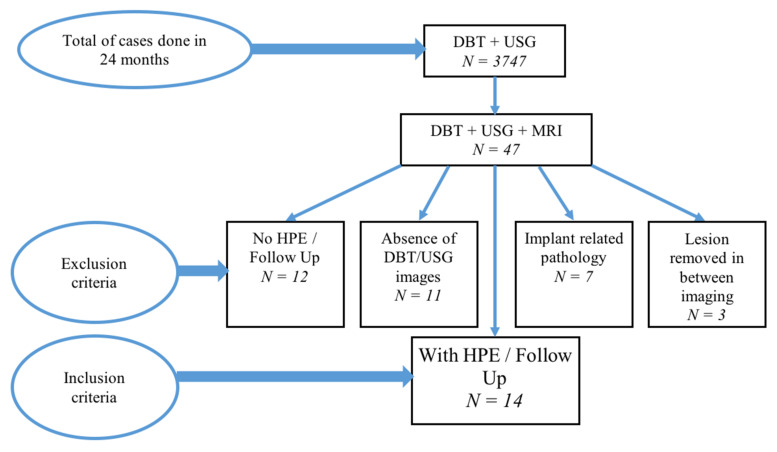
Patient selection in present study.

**Figure 2 ijerph-19-00759-f002:**
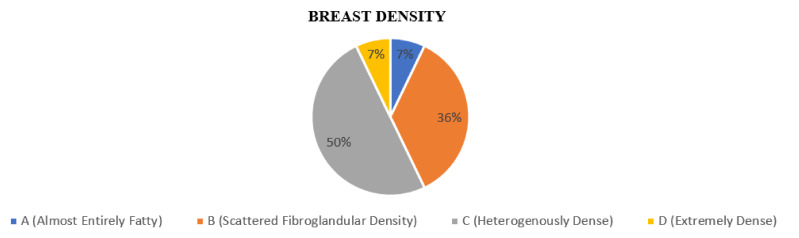
Proportion according to breast density on DBT.

**Table 1 ijerph-19-00759-t001:** Demographic data of patient in our study.

No. of Patient	Age (Year)	Ethnicity	Symptom	Duration Interval between Two Imaging Modalities
1.	72	Indian	X	≤1 month
2.	56	Malay	X	≤1 month
3.	65	Malay	√	≤1 month
4.	58	Chinese	X	>3 months
5.	58	Malay	X	>3 months
6.	60	Malay	√	>3 months
7.	54	Malay	X	>3 months
8.	78	Chinese	√	>1 month to ≤3 months
9.	42	Malay	√	Same day
10.	36	Malay	√	>1 month to ≤3 months
11.	40	Chinese	√	>1 month to ≤3 months
12.	65	Malay	X	≤1 month
13.	42	Malay	√	>3 months
14.	46	Malay	√	≤1 month

Symptom: “X” = Asymptomatic, “√” = Symptomatic.

**Table 2 ijerph-19-00759-t002:** Summary of lesions detected in the present study.

No.	Age	Breast Composition (BI-RADS)	BI-RADS Category			Follow Up DBT (BI-RADS)	CONCLUSION (BIOPSY/FOLLOW UP)
			DBT	DBT + USG	MRI		
1.	72	b	4	4	4	N/A	ILC
2.	56	b	5	5	5	N/A	IDC
3.	56	b	1	1	5	N/A	IDC
4.	65	c	4	5	5	N/A	IDC
5.	65	c	1	1	2	1	Stable lesion
6.	65	c	1	1	2	1	Stable lesion
7.	65	c	1	1	2	1	Stable lesion
8.	65	c	3	2	2	2	Benign breast tissue
9.	58	c	1	2	2	1	Stable lesion
10.	58	c	1	2	2	1	Stable lesion
11.	58	c	1	1	2	1	Fibroglandular tissue
12.	58	c	3	2	2	2	Scar
13.	58	c	4	4	2	2	Radiotherapy changes
14.	60	c	2	2	2	2	Stable lesion
15.	54	a	4	5	5	N/A	IDC
16.	78	c	1	3	4	N/A	Benign breast tissue
17.	42	c	4	4	4	N/A	Benign breast tissue
18.	36	d	5	5	5	N/A	IDC
19.	36	d	1	2	3	N/A	Benign breast tissue
20.	40	c	4	5	5	N/A	ILC
21.	65	c	5	5	5	N/A	ILC
22.	65	c	1	1	2	1	Resolved
23.	42	c	2	2	2	N/A	Fibroadenoma
24.	46	c	4	5	5	N/A	ILC

N/A = not available, IDC = invasive ductal carcinoma, ILC = invasive lobular carcinoma.

**Table 3 ijerph-19-00759-t003:** Comparison of MRI breast, DBT + USG and DBT alone compared to HPE result or follow up DBT.

Type of Lesion	Histopathology/Follow Up DBT		TOTAL
	Benign	Malignant	
DBT			
Benign	11	1	12
Malignant	4	8	12
DBT + USG			
Benign	12	1	13
Malignant	3	8	11
MRI breast			
Benign	10	0	10
Malignant	5	9	14
TOTAL	15	9	24

**Table 4 ijerph-19-00759-t004:** Accuracy, sensitivity, specificity, PPV and NPV for DBT, DBT + USG and MRI.

	Accuracy (%)	Sensitivity (%)	Specificity (%)	PPV (%)	NPV (%)
DBT	79.1	88.9	73.3	66.7	91.6
DBT + USG	83.3	88.9	80	72.7	92.3
MRI	79.1	100	66.7	64.3	100

**Table 5 ijerph-19-00759-t005:** *p* values of DBT, DBT + USG and MRI according to chi-square.

Modalities	Ch^2^	df	*p*-Value
DBT	8.711	1	0.003 *
DBT + USG	10.752	1	0.001 *
MRI	10.286	1	0.001 *

Notes: * indicates significant associations (*p* < 0.005).

**Table 6 ijerph-19-00759-t006:** Comparison in components of the present study with other studies.

Data Source	Year	No. of Lesion/Patient	DBT	DBT + USG	DM + DBT + USG	MRI
Present study	2021	24 lesions	√	√		√
Sudhir et al.	2021	166 lesions	√	√		
Comstock et al.	2020	1444 patients	√			√
Dibble et al.	2019	3183 breasts		√		
Saxena et al.	2019	50 patients	√			√
Kamal et al.	2016	103 lesions	√			√
Roganovic et al.	2015	57 lesions	√			√
Mariscotti et al.	2014	200 patients	√		√	√

## Data Availability

The datasets generated during and/or analysed during the current study are not publicly available due to patients’ privacy but are available from the corresponding author on resonable request.

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
