# Peer review of "Comparative Study of Digital Breast Tomosynthesis (DBT) with and without Ultrasound versus Breast Magnetic Resonance Imaging (MRI) in Detecting Breast Lesion"

_ijerph, 2022, doi:10.3390/ijerph19020759_

Round 1
Reviewer 1 Report
Overall:
This paper presents a retrospective study to evaluate the use of DBT alone, DBT with US vs. Breast MRI for the diagnosis of breast lesions.
This type of evaluation is very welcome for the scientific world as well as governments and decision-makers.
Some issues with this work are:
- Is not clear how many radiologists and how many years of experience they have.
- The variability in the examination interval might be a factor to be considered and could jeopardize the evaluation and conclusions. Images taken on the same day might be very different from those taken in more than 3 months of interval.
- The dataset is extremely limited, only 14 patients. Moreover, there are several other limiting factors such as ethnicity and breast density. All this can influence a lot of the conclusions if you have such a small dataset.
Besides some necessary clarification, I suggest the authors try to improve the number of patients considered.
Author Response
"Please see the attachment."
Overall:
This paper presents a retrospective study to evaluate the use of DBT alone, DBT with US vs. Breast MRI for the diagnosis of breast lesions.
This type of evaluation is very welcome for the scientific world as well as governments and decision-makers.
We thank the reviewer for his/her kind words.
Some issues with this work are:
- Is not clear how many radiologists and how many years of experience they have: Has been added into text as suggested.:5 radiologists with less than 2 years experience.
- The variability in the examination interval might be a factor to be considered and could jeopardize the evaluation and conclusions. Images taken on the same day might be very different from those taken in more than 3 months of interval. Thank you for the feedback. We will add this into our limitations
- The dataset is extremely limited, only 14 patients. Moreover, there are several other limiting factors such as ethnicity and breast density. All this can influence a lot of the conclusions if you have such a small dataset.
Besides some necessary clarification, I suggest the authors try to improve the number of patients considered. We appreciate the input. Although 3747 patients underwent DBT during the study period, only 14 fulfilled the stringent inclusion criteria. Since the study design is retrospective, the actual imaging was done a few years back. If we change the study design, the population may no longer be the same.

Reviewer 2 Report
This article is interesting, although there are similar comparisons in other countries. Despite the small number of female patients (?), the results should be published. The discussion should be improved, and it should not include general information on the topic. This should be in the Introduction.
Please take into account my comments.
Line 10
I do not agree that
„Ultrasonography (USG) is commonly used to 10 supplement mammography.” This examination is also performed as an independent breast examination.
Line 26
Breast cancer is not just a cancer that affects women. It is estimated that about 1% of all malignant breast tumors are diagnosed in men. The authors should take note of this.
Line 27
Citation [1] does not provide current data, please change to newer data. In addition, in many publications, even those from 10 years ago, breast cancer (among women) ranks first.
Line 30
Why the Malaysian guidelines? Please also cite global guidelines, the article in the title is not that in Malaysia.
Lines 37 and 62-64 likewise
„… in women with dense breasts…
Probably the authors were interested in women with denser connective tissue fibers, which dominate in the breasts of young women.
Line 77
These are not all three methods of diagnosing breast cancer yet: breast self-egzamination, galactography (ductography), scyntymammography, Miralum test, PET, Termography (DITI), magnetic resonance elastography; ultrasound elastography, Computed Tomography Laser Mammography (CTLM) or Diffuse Optical Imaging
Line 89
change from, (if only women, men can also get breast cancer)
Forty-seven patients out of the 3747…
change to
Forty-seven female patients out of the 3747…
Or indicate that only women were tested.
In Material;
please add the number of ethnic groups and the age of the individuals
Line 163
„Chart 3. Distribution of breast densities according to age group.”
With such a small number, this figure is redundant, delete.
Line 182, Table 5
insert the correct symbol Ch2
In Metods;
Please enter that there were histopathological analyzes
Lines 185-197; 215-227
This is the information that should be included in the Introduction. Additionally, some of them are repeated with the already existing ones in the introduction.
Lines 198-199
Also, the statement that there are several similar studies (with citation) should be referred to in the introduction and then in the discussion.
Line 292
What radiographs, incomprehensible sentence. Perhaps it relates to classical digital mammography?
I propose to include two publications in the Discussion : https://doi.org/10.1148/radiol.14141233; https://doi.org/10.1177/20584601211063746
Author Response
"Please see the attachment."
